# The ALGOVUE Clinical Trial: Effects of the Daily Consumption of Eggs Enriched with Lutein and Docosahexaenoic Acid on Plasma Composition and Macular Pigment Optical Density

**DOI:** 10.3390/nu13103347

**Published:** 2021-09-24

**Authors:** Coralie Schnebelen-Berthier, Niyazi Acar, Emilie Simon, Clémentine Thabuis, Anne Bourdillon, Adeline Mathiaud, Luc Dauchet, Cécile Delcourt, Pascale Benlian, Martine Crochet, Sabine Defoort, Anne Tailleux, Bart Staels, Lionel Bretillon, Jean-Michel Lecerf

**Affiliations:** 1Service Nutrition & Activité Physique, Institut Pasteur de Lille, F-59019 Lille, France; emilie.simon@live.fr (E.S.); jean-michel.lecerf@pasteur-lille.fr (J.-M.L.); 2Centre des Sciences du Goût et de l’Alimentation, Agrosup Dijon, CNRS, INRA, Université Bourgogne Franche-Comté, F-21000 Dijon, France; niyazi.acar@inrae.fr (N.A.); lionel.bretillon@inrae.fr (L.B.); 3ROQUETTE, F-62080 Lestrem, France; clementine.thabuis@roquette.com; 4MIXSCIENCE, Centre d’Affaires Odyssée, ZAC Cicé Blossac, F-35172 Bruz, France; anne.bourdillon@mixscience.eu (A.B.); adeline.mathiaud@mixscience.eu (A.M.); 5CNRS, INSERM, CHU de Lille, Institut Pasteur de Lille, UMR 1283-EGID, “Integrative Genomics and Modelling of Metabolic Diseases”, Université de Lille, F-59045 Lille, France; luc.dauchet@pasteur-lille.fr (L.D.); pascale.benlian@univ-lille.fr (P.B.); anne.tailleux@univ-lille2.fr (A.T.); bart.Staels@pasteur-lille.fr (B.S.); 6“Bordeaux Population Health Research Center” (BPH), INSERM, UMR1219, “Lifelong Exposures, Health and Ageing” (LEHA) Group, Université de Bordeaux, F-33076 Bordeaux, France; cecile.delcourt@u-bordeaux.fr; 7CHU Lille, Service d’Exploration de la Vision et Neuro-Ophtalmologie, F-59000 Lille, France; m.crochet@wanadoo.fr (M.C.); Sabine.DEFOORT@chru-lille.fr (S.D.)

**Keywords:** enriched egg, lutein, docosahexaenoic acid (DHA), age-related macular degeneration (AMD), macular pigment, microalgae, xanthophylls

## Abstract

Background. Carotenoids and docosahexaenoic acid (DHA) were identified as essential components for eye health and are both naturally present in eggs. Objective. We aimed to evaluate the effect of the daily consumption of two eggs enriched with lutein/zeaxanthin and DHA on macular pigment optical density (MPOD) and on circulating xanthophyll and fatty acid concentrations in healthy participants. Methods. Ninety-nine healthy volunteers consumed either two standard eggs or two enriched eggs per day for 4 months. MPOD was measured at baseline (V0) and at follow-up (V4) using a modified confocal scanning laser ophthalmoscope (primary outcome). Blood samples were collected to determine total plasma and lipoprotein fatty acids and lutein/zeaxanthin compositions at V0 and V4 (secondary outcomes). Results. A slight but significant increase in MPOD was observed for all study participants consuming two eggs per day for 4 months at all eccentricities (0.5°, 1°, 2°, and 4°). Plasma and lipoprotein lutein, zeaxanthin, and DHA concentrations significantly increased in both groups but were greater in the enriched group (for the enriched group (V0 vs. V4): lutein, 167 vs. 369 ng/mL; zeaxanthin, 17.7 vs. 29.2 ng/mL; DHA, 1.89 vs. 2.56% of total fatty acids). Interestingly, lutein from high-density lipoprotein (HDL) was strongly correlated with MPOD at 0.5 and 1° eccentricities (rho = 0.385, *p* = 0.008, and rho = 0.461, *p* = 0.001, respectively). Conclusions. MPOD was slightly increased in both groups. Lutein, zeaxanthin, and DHA plasma concentrations were strongly enhanced in the enriched group compared with the standard group. A significant correlation was found between MPOD level and lutein concentration in HDL.

## 1. Introduction

Age-related macular degeneration (AMD) is a leading cause of irreversible vision loss affecting people over age 50 in Western countries [1]. Although the understanding of AMD pathogenesis is progressing rapidly, this disease is not totally understood. Epidemiological studies have reported that the highest dietary intakes of vitamins C and E, zinc, lutein, zeaxanthin, and long chain omega-3 polyunsaturated fatty acids (PUFAs), such as docosahexaenoic acid (DHA), are associated with the greatest reduction in the risk of developing early [2,3,4,5] and advanced AMD [6,7].

As primary constituents of macular pigment, carotenoids lutein and zeaxanthin are believed to play crucial roles in maintaining the morphological and functional integrity of the retina and, more specifically, of the macula [8]. Carotenoids have antioxidant properties and act as filters for blue light [9,10]. They may thus protect the macula against light-induced oxidative damage, which has been implicated in the pathogenesis of AMD [11]. DHA is a fatty acid found in high quantities in the retina, especially in photoreceptor outer segments [12]. In vitro and in vivo studies on animals have shown that DHA plays multiple functions in photoreceptors, including a structural role in membranes and specific interactions with rhodopsin protein, which suggests that DHA influences visual transduction processes [13]. Moreover, DHA is the precursor of oxygenated mediators that display anti-inflammatory and anti-apoptotic properties [14]. The results of animal studies of chronic deficiencies in DHA are in concordance with the crucial roles played by this fatty acid in retinal functionality and retinal cell integrity [15,16,17].

Previous observations in humans and animals have shown that xanthophylls such as lutein and zeaxanthin are delivered to tissues after being transported primarily by high-density lipoproteins (HDLs) [18,19]. Given that dietary DHA increases plasmatic HDL concentrations [20,21], clinical interventions hypothesizing that DHA would promote xanthophyll transport to the retina have been carried out [22,23,24]. Thanks to its design, only the study of Johnson and collaborators definitely demonstrated that adding DHA to a lutein-rich supplement enhances plasma lutein concentration and macular pigment optical density (MPOD), which is an indirect estimate of xanthophyll concentration in the retina. However, while the authors measured the proportions of the different lipoprotein subclasses, their effective enrichment in DHA and lutein was not investigated.

The objective of the present work was to evaluate the effects of adding DHA to lutein and zeaxanthin supplementation on MPOD and on the composition of total plasma and plasma lipoproteins in healthy participants. Moreover, we hypothesized that low amounts of xanthophylls and DHA could increase MPOD using eggs, which are known to increase nutrient bioavailability due to their phospholipid content [25], as a vector for supplementation. Finally, the last novelty of this study is the use of *Schizochitrium* sp. microalgae and alfalfa protein concentrate as a source of DHA and lutein, respectively, to enrich hens’ eggs to increase lutein and DHA intake and thus potentially improve macular pigment density.

## 2. Materials and Methods

### 2.1. Ethics Statements and Selection of Participants

All the participants gave their written consent after being informed about the purpose, methodology, and potential risks of the study. This experiment was conducted in the clinical center of the Institut Pasteur de Lille in accordance with the guidelines of the Declaration of Helsinki of 1975 as revised in 1983, and the experiment was approved by the local ethics committee (CPP Nord Ouest IV, CHU Lille, Lille, France, study number: 11/31, accepted on 3 May 2011). This study was registered at the French agency for the safety of health products (Agence Nationale de Sécurité du Médicament et des Produits de Santé, ANSM) under the number 2011-A00364-37 and at clinicaltrials.gov as NCT01480700 (available online: https://clinicaltrials.gov/ accessed on 23 September 2021).

One hundred twenty-five nonsmoking healthy men and women with a BMI ≤ 30 kg/m^2^ aged 18 to 55 years were screened, and ninety-nine were enrolled in the study (Figure 1). Participants with hypolipemic treatment, total cholesterol > 2.5 g/L, triglycerides > 2.0 g/L, diabetes, cardiovascular disease, currently evolving disease (cancer, neurodegenerative disease, etc.), or ocular disorders (maculopathy, optical neuropathy, cataract, anomalies in color vision, and corrected visual acuity of less than 8/10) were not included in the study. Other exclusion criteria were the high consumption of carotenoids (more than 4 servings per week of foods rich in carotenoids), omega-3 fatty acids, or phytosterols (through dietary supplement or diet—e.g., consumption of oily fish 4 times a week or more, regular consumption of Danacol^®^-Danone, Rueil-Malmaison, France-or pro-activ’ margarine^®^-Unilever, Asnières sur Seine, France-) during the last 3 months, and known hypersensitivity or allergy to eggs.

### 2.2. Study Design

#### 2.2.1. Study Type and Objectives

This study was a monocenter, double-blind (participants and experimental staff were blind), randomized trial designed to investigate the effect of the consumption of eggs enriched with lutein and DHA on macular pigment optical density (primary outcome). Blood concentrations of xanthophylls and fatty acids were also investigated as secondary outcomes. The random allocation sequence was generated by a biostatistician using blocks of 4 and considering both egg composition and first eye performing MPOD analyses. Medical staff enrolled the participants and assigned their randomization in order of their arrival (signed consent).

#### 2.2.2. Eggs

Study participants were asked to consume 2 eggs per day for 4 months. These eggs were either standard eggs or enriched eggs with lutein and DHA. To improve compliance and to favor meal diversification, participants were provided with a book containing more than 100 recipes using eggs, which was especially developed for the study by a dietician. No further recommendation was made on the way to consume the eggs.

The eggs were obtained from laying hens fed either a standard diet or a diet enriched with lutein and DHA at the experimental farm of MIXSCIENCE (Sourches, France). Lutein in the hens’ diet originated from alfalfa protein concentrate, whereas DHA was provided by *Schizochytrium* sp. microalgae produced by ROQUETTE (Lestrem, France). The lutein and DHA contents per egg were 0.12 mg and 37.6 mg for standard eggs and 0.96 mg and 134.4 mg for enriched eggs, respectively. The carotenoid, lipophilic vitamin, and polyunsaturated fatty acid compositions of the eggs are described in Table 1. The composition of standard eggs in this study was comparable with that of other standard eggs in the French market [26,27].

#### 2.2.3. Dietary Recommendations

Before starting the study, the participants were asked to record in a food intake questionnaire all food ingested over seven days to determine dietary patterns and estimate energy and macronutrient and micronutrient intakes. The same questionnaire was completed by the participants during the week preceding the end of the study. The food intake questionnaire was completed by participants each day (or after each meal if possible) for 7 days (paper completion). Each food intake questionnaire was verified by a trained dietician using suvimax photography. Participants were asked not to change their dietary habits during the study (except for eggs consumption), to consume no egg other than those proposed in the study, and to remove all foods rich in lutein (cabbage, spinach, flaxseed, etc.), phytosterols (Danacol^®^ -Danone, Rueil-Malmaison, France-, pro-activ’^®^ margarine—Unilever, Asnières sur Seine, France- etc.), and DHA (oily fish such as sardine, mackerel, salmon, etc.) from their diet. No specific egg cooking guidelines were given, as egg preparation methods had no effects on egg lutein and DHA availability [26,27]. However, a book especially developed for the study with cooking recipes and advice was given to participants in the two groups.

### 2.3. Blood Samples

Blood sample collection was conducted after overnight fasting before the start (V0) and at the end of the 4-month intervention period (V4). Blood samples were collected into tubes without additive for serum lipid analyses (V0 and V4) and into EDTA-coated tubes for the separation of plasma, nucleated cells, and erythrocytes (V0 and V4). Samples destined for lipid analyses were maintained at room temperature for at least 30 min to allow clotting and were then centrifuged at a low speed of 1300× *g* for 10 min at 20 °C to isolate serum. Serum total cholesterol, HDL cholesterol, and triglyceride concentrations were determined by automaton (AU400, Olympus Medical, Hamburg, Germany) using an enzymatic colorimetric method. Serum LDL concentrations were determined according to the Friedewald calculation. EDTA-coated tubes were centrifuged at 1800× *g* for 10 min at 4 °C to isolate plasma, nucleated cells, and erythrocytes. These samples were stored at −80 °C until further analyses. LDL and HDL fractions were isolated from plasma by sequential ultracentrifugation [28] using a Beckman ultracentrifuge (TLA-100.4 rotor, 543,000× *g* (100,000 rpm), +10 °C). Fraction protein content was measured using a Lowry assay.

### 2.4. Determination of Plasma Lipoprotein and Erythrocyte Membrane Fatty Acids

All chemical reagents were purchased from Sigma-Aldrich (St Quentin, Fallavier, France), and chloroform and methanol were purchased from SDS (Peypin, France). Total lipids were extracted from total plasma, erythrocytes, and LDL- and HDL-fractions according to Moilanen and Nikkari [29]. Total phospholipids from total plasma, erythrocyte membranes, and LDL- and HDL fractions were transmethylated using boron trifluoride in methanol according to Morrison and Smith [30]. Fatty acid methyl esters (FAMEs) were extracted with hexane and analyzed by gas chromatography on a Hewlett Packard Model 5890 gas chromatograph (Hewlett-Packard, Palo Alto, CA, USA) using a CPSIL-88 column (100 m × 0.25 mm i.d., film thickness 0.20 µm; Varian, Les Ulis, France) equipped with a flame ionization detector. Hydrogen was used as the carrier gas (inlet pressure 210 kPa). The oven temperature was held at 60 °C for 5 min, increased to 165 °C at 15 °C/min and held for 1 min, and then to 225 °C at 2 °C/min, and finally held at 225 °C for 17 min. The injector and the detector were maintained at 250 °C. FAMEs were identified by comparison with commercial and synthetic standards. The data were processed using EZChrom Elite software (Agilent Technologies, Massy, France) and reported as a percentage (mole %) of the total fatty acids.

### 2.5. Determination of Total Plasma and Lipoprotein Lutein and Zeaxanthin Concentrations

The concentrations of lutein and zeaxanthin in total plasma and LDL- and HDL-fractions were determined according to previously described procedures [31,32]. All further processes were carried out under yellow light. After adding echinenone as an internal standard, carotenoids were extracted twice with hexane. The extracts were dried under a stream of nitrogen and then dissolved in a known volume of mobile phase (see below) for HPLC analysis.

The HPLC system consisted of a solvent degasser (model DG-1580-53, Jasco, Bouguenais, France), a pump set up in isocratic mode (model PU-1580 and LG-1580-02, Jasco, Bouguenais, France), an autosampler (model Spectra System AS300, Thermo Finnigan, Villebon Sur Yvette, France) maintaining samples at 15 °C, a Nucleosil C18 column (25 mm × 4.6 mm i.d., film thickness 5 µM, Thermo Finnigan, Villebon Sur Yvette, France), and a VIDAK C18 column (25 mm × 4.6 mm i.d., film thickness 5 µM, GRACE, Altech France, Epernon, France), both maintained at 37 °C by a column heater, a guard column containing a C18 phase (10 mm × 2 mm i.d., film thickness 5 µM), a light scattering detector (model MD-1510 Multiwavelength detector, Jasco, Bouguenais, France), and computer data system (Borwin PDA, Jasco, Bouguenais, France). The mobile phase for the isocratic separation was 70% acetonitrile, 15% methanol containing 50 mM ammonium acetate, 5% water, and 10% dichloromethane, and the flow rate was 2 mL/min. Carotenoids were measured by absorbance at 450 nm and a wavelength scan from 200 to 650 nm. Their concentrations were determined from peak areas using internal and external standards. Linear calibration curves were prepared consisting of multiple concentrations of echinenone.

### 2.6. Measurement of Macular Pigment Optical Density (MPOD)

All participants underwent a comprehensive ocular examination. Their pupils were dilated with eye drops containing 0.5% tropicamide and 2.5% phenylephrine. MPOD was measured using a modified confocal scanning laser ophthalmoscope (SLO, model mpHRA, Heidelberg Engineering, Heidelberg, Germany) by means of autofluorescence images [33,34]. Each participant was positioned in front of the SLO camera and was instructed to look straight ahead. After focusing the SLO on the macular area, image sequences were captured at 488 nm (maximal absorption) and 514 nm (minimal absorption) at least 30 s after retinal bleaching. MPOD was recorded at eccentricities of 0.5, 1.0, 2.0, and 4.0 degrees, and mean MPOD values were calculated for each using the software provided by the manufacturer. MPOD was expressed as optical density units (DU).

### 2.7. Statistical Analyses

Normality of variables was tested by the Shapiro–Wilk test. Log transformations were performed when the distribution was skewed to obtain normally distributed variables. Comparison of means was done using the *t* test. The change between V0 and V4 was tested using a paired *t* test. The effect of the supplementation was evaluated by comparing the mean change in each group using the *t* test. Sensitivity analysis was performed with a general linear model adjusted for age, triglycerides, total cholesterol, and LDL cholesterol. To have a single test for all eccentricities, we analyzed the four eccentricities in one mixed model. We used the lmer function of the R package lme4. MPOD was the dependent variable. The visits (V0, V4) and eccentricity were included in the model with a random slope and intercept. Spearman correlations were calculated between DHA, lutein, zeaxanthin, and MPOD. All analyses were performed using R software (available online: http://www.R-project.org/ accessed on 23 September 2021).

### 2.8. Sample Size Calculation

The sample size calculation of our trial was based on the MPOD measurement reported in the study of Vishwanathan et al. (2009) [35]. In this study, there was a nonsignificant increase in macular pigment at 0.5° (0.06 units, SD 0.13) following consumption of 4 eggs per day for 5 weeks (providing on average 1886 μg of lutein + zeaxanthin per day, *n* = 37) (data from the figures). Based on these results, we expected a small increase in the value of the macular pigment (0.002 units) in the standard group and a more consequent increase in the value of the macular pigment (0.08 units) for the enriched group. Therefore, estimating an increase in macular pigment value of ∆ = 0.08 units in the group consuming enriched eggs, a standard deviation of this increase of *σ* = 0.13, setting *α* = 5% (*z*_1−*α*/2_ = 1.96) and *β* = 20% (*z*_1−*β*_ = 0.8416) and using the formula for determining the minimum number of participants in a bilateral situation, we obtained the following:$$n=2\times {\left[\frac{\left(z_{1-\frac{\alpha }{2}}+z_{1-\beta }\right)}{\Delta }\times \sigma \right]}^{2}=2\;\times {\left[\frac{\left(1.96+0.8416\right)}{0.08}\times 0.13\right]}^{2}=45$$

To compensate for possible protocol deviations and to increase statistical power, we chose to enroll 50 participants per group.

## 3. Results

### 3.1. Baseline Characteristics

The baseline characteristics of the participants are presented in Table 2. For the two groups, clinical and biological parameters at baseline were in the usual range for age and sex. There was no difference between the two groups of participants regarding sex, BMI, blood pressure, heart rate, HDL cholesterol, LDL cholesterol/HDL cholesterol ratio, MPOD at any degree of eccentricity, or blood carotenoid and DHA concentrations. However, age, plasma triglycerides, total cholesterol, and LDL cholesterol concentrations were higher in participants from the enriched group than in those from the standard group. Because a sensitivity analysis performed with a general linear model adjusted on these four parameters did not show any effect on the significance of the results, these differences in age, triglycerides, total cholesterol, and LDL cholesterol concentrations between the groups of participants at baseline were not taken into account.

Data from food intake questionnaires at baseline did not show any statistically significant difference in total energy, protein, carbohydrate, total fat, saturated fatty acid, monounsaturated fatty acid, polyunsaturated fatty acid, DHA, cholesterol, fiber, carotenoid, vitamin (A, E, D), or micronutrient (such as sodium, calcium, magnesium, potassium, iron, and selenium) intake between the two groups of participants (Appendix A).

### 3.2. Adverse Events and Compliance

No adverse events related to the study were observed or reported by participants. During the four-month intervention, the participants consumed an average of 13.5 eggs (mean ± 0.7(SD)) per week instead of 14 (13.50 ± 0.65 in control group, 13.55 ± 0.72 in enriched group), corresponding to an adherence rate of 93.6%.

### 3.3. Effect of Lutein- and DHA-Enriched Egg Consumption on Clinical Parameters

Even if the change was low (+1.3%), BMI was increased 4 months after egg consumption in the enriched group only. Except for this parameter, no difference in the mean change in clinical parameters between V0 and V4 was observed in the standard vs. enriched groups. We also focused the description of the results on the total sample (Table 3).

No effect was observed on heart rate or triglyceride plasma concentration after 4 months of egg consumption. However, total cholesterol (1.81 g/L at V0 vs. 1.90 g/L at V4, *p* = 6.00 × 10^−5^), HDL cholesterol (0.57 g/L at V0 vs. 0.59 g/L at V4, *p* = 0.013), and LDL cholesterol (1.09 g/L at V0 vs. 1.15 g/L at V4, *p* = 0.0002) significantly increased for all participants after 4 months of daily consumption of two eggs per day. The LDL cholesterol/HDL cholesterol ratio also tended to increase (*p* = 0.054). Blood pressure decreased for all participants consuming two eggs per day for 4 months (120.7 mmHg at V0 vs. 118.4 mmHg at V4).

### 3.4. Effect of Eggs Consumption on MPOD

Compared with baseline, MPOD increased slightly but significantly at all eccentricities in the total sample after 4 months of egg consumption (*p* = 0.03 at 0.5°; *p* = 0.00008 at 1°; *p* = 0.0005 at 2°; and *p* = 0.0028 at 4°; Table 4). Similar results were observed in participants consuming standard eggs, except at 0.5° eccentricity. In participants consuming enriched eggs, a significant increase in MPOD at 1° of eccentricity was only observed after 4 months of egg consumption (*p* = 0.025). However, a comparison of the mean change in MPOD at all eccentricities between V0 and V4 in the enriched group vs. the standard group did not show any difference (+2%, *p* = 0.98 at 0.5°; +3%, *p* = 0.8 at 1°; +1%, *p* = 0.4 at 2°; and −3%, 0.19 at 4°). In the mixed model including all eccentricities, MPOD significantly increased between V0 and V4 in the whole sample (*p* < 0.002) and in the standard group (*p* < 0.004) and tended to be significant in the enriched group (*p* = 0.06). The impact of the group on the MPOD change was not significant (*p* = 0.68).

### 3.5. Effect of Lutein- and DHA-Enriched Egg Consumption on Plasma and Lipoprotein Content in Lutein and Zeaxanthin

The consumption of two eggs per day for 4 months induced a significant increase in total plasma lutein concentrations in participants from the standard group (from 154 ng/mL to 178 ng/mL (+15%), *p* = 0.001) and the enriched group (from 167 ng/mL to 369 ng/mL (+120%), *p* = 3.13 × 10^−16^; Table 4). The mean changes in total plasma lutein concentrations between V0 and V4 were significantly greater in participants consuming the enriched eggs than in participants consuming the standard eggs (*p* = 5.26 × 10^−13^). Similar results were observed for plasmatic zeaxanthin concentrations, as they increased from 17.6 ng/mL to 22.7 ng/mL (+29%) in the standard group (*p* = 7.13 × 10^−6^) and from 17.7 ng/mL to 29.2 ng/mL (+65%) in the enriched group (*p* = 3.38 × 10^−13^). For lutein, the mean changes in total plasma zeaxanthin concentrations between V0 and V4 were also greater in participants consuming eggs enriched with xanthophylls (*p* = 0.003).

The increase in the concentrations of lutein and zeaxanthin in total plasma of all participants was likely to be the consequence of the enrichment of HDL lipoproteins in these carotenoids (from 87.4 ng/mL to 99.3 ng/mL for lutein in standard group, *p* = 0.02; from 91.4 ng/mL to 179.3 ng/mL for lutein in enriched group, *p* = 4.31 × 10^−11^; from 10.4 ng/mL to 13.3 ng/mL for zeaxanthin in standard group, *p* = 0.0005; from 10.0 ng/mL to 15.0 ng/mL for lutein in enriched group, *p* = 1.0 × 10^−9^; Table 4). In participants consuming the enriched eggs, the enrichment of HDL in lutein and zeaxanthin was even larger than in the standard group (*p* = 1.30 × 10^−8^ for lutein and *p* = 0.01 for zeaxanthin when comparing the change of both groups). In the enriched group, a significant increase in lutein and zeaxanthin concentrations in LDL lipoproteins was also observed after four months of study (from 85.6 ng/mL at V0 to 182.5 ng/mL at V4 for lutein, *p* = 1.75 × 10^−12^; from 9.5 ng/mL at V0 to 14.4 ng/mL at V4 for zeaxanthin, *p* = 1.50 × 10^−6^). However, this increase was not observed for the standard group.

By checking for associations between MPOD and xanthophyll plasma concentrations at the end of the study, we identified a strong correlation in participants consuming enriched eggs between HDL concentrations of lutein and MPOD at 0.5° and 1° of eccentricities, where this carotenoid is most concentrated in the human retina [36,37] (rho = 0.385, *p* = 0.008 and rho = 0.461, *p* = 0.001, respectively; Figure 2). Similar results were observed at 2° and 4° eccentricities (rho = 0.479, *p* = 0.001 and rho = 0.474, *p* = 0.001, respectively; data not shown). However, no significant association was observed between MPOD and HDL concentrations of lutein in the standard group. In participants consuming standard eggs, statistically significant associations were observed between LDL concentrations of lutein and MPOD at 2° and 4° of eccentricities (rho = 0.329, *p* = 0.029 and rho = 0.338, *p* = 0.025, respectively; data not shown). No relevant association was observed between MPOD and total plasma, HDL, and LDL zeaxanthin concentrations.

### 3.6. Effect of Lutein- and DHA-Enriched Egg Consumption on Total Plasma, Red Blood Cells, and Lipoprotein Concentrations of DHA

The concentrations of DHA significantly increased in the total plasma and in the red blood cells of participants from both groups (Table 4). For plasma lutein, the increase in DHA concentrations was more substantial in participants consuming enriched eggs (plasma: from 1.94% to 2.07% of total fatty acids in standard group, *p* = 0.03; from 1.89% to 2.56% of total fatty acids in the enriched group, *p* = 1.00 × 10^−11^; red blood cells: from 1.84% to 2.97% of total fatty acids in standard group, *p* = 3.11 × 10^−7^; from 2.04% to 4.10% of total fatty acids in the enriched group, *p* = 2.71 × 10^−15^). The large increase in plasma and red blood cell DHA concentrations in participants from the enriched group was associated with a strong elevation of DHA concentrations in HDL and LDL fractions (from 2.47% to 3.18% of total fatty acids in HDL, *p* = 2.62 × 10^−9^; from 1.56% to 1.99% of total fatty acids in LDL, *p* = 1.49 × 10^−5)^. No variation in DHA concentrations was observed in HDL and LDL fractions of participants consuming standard eggs.

## 4. Discussion

The objective of this study was to determine whether the daily consumption of two eggs enriched with lutein and zeaxanthin (from alfalfa protein concentrate) and DHA (from *Schizochytrium* sp. microalgae) would affect macular pigment optical density and blood concentrations of xanthophylls and fatty acids in healthy human participants.

In this trial, we decided to use low amounts of xanthophylls and DHA and to deliver them through a dietary vector that may have given them excellent bioavailability. Indeed, it is known that the bioavailability of lipophilic nutrients such as lutein, zeaxanthin, and fatty acids depends on the matrix in which they are incorporated [25,38]. Several studies have shown that serum concentrations of xanthophylls and MPOD increase more when present in a lipid matrix such as egg yolk when compared with other food products or pure preparations as dietary supplements (reviewed by Thurnham, 2007 [39]). Egg yolk is also known to be rich in lipids and to contain approximately one-third of these lipids in the phospholipid form, whereas triglycerides represent the major lipid class in food products and food supplements. It is now well accepted that PUFAs provided by dietary phospholipids have a higher intestinal bioavailability than those delivered by triglycerides (reviewed by Schuchardt and Hahn, 2013 [25]).

As a consequence, whereas the majority of studies using food supplements have used daily doses of 6 mg of lutein and 2 mg of zeaxanthin, we gave only 1.90 mg and 0.20 mg of these xanthophylls per day, respectively, and only 269 mg of DHA per day, which is much lower than previous intervention trials in ophthalmology [40,41]. Because the standard eggs naturally contained lutein, zeaxanthin, and DHA (daily doses of 0.24 mg, 0.13 mg, and 75 mg, respectively), the concentrations of these molecules were significantly increased in the total plasma and HDL fraction for lutein and zeaxanthin and in the total plasma for DHA in participants from the standard group. In participants consuming the enriched eggs, the increase in lutein, zeaxanthin, and DHA was even greater. These results are in accordance with the Blesso and collaborators’ study showing a significant enrichment of isolated HDL and LDL fractions in lutein and zeaxanthin in participants consuming three whole eggs per day [42]. It is well known that lutein and zeaxanthin are mostly associated with HDL, followed by LDL and VLDL [43]. In our study, plasma lutein incorporation was 2.2-fold higher after 4 months of enriched egg consumption compared with baseline. Comparable results were observed by Kelly and collaborators since a 2.3-fold higher concentration was observed in serum lutein after daily enriched egg consumption (ratio lutein/mesozeaxanthin 1:1) for 2 months [44]. This increase was higher than for lutein supplement; Obana and collaborators showed that 10 mg of lutein supplement from marigold oleoresin increased lutein plasma concentration by 1.7 times compared with baseline after 6 months of supplementation in 36 participants [45]. Similarly, Bone and collaborators observed approximately 2-fold higher carotenoid serum concentration with Lumega-Z (a lipid-based micronized liquid medical food) containing 28 mg of lutein, zeaxanthin, and meso-zeaxanthin after 12 weeks of supplementation in healthy participants [46]. It can then be hypothesized from these data that lutein from eggs was more incorporated in plasma than lutein from the supplement. Chung and collaborators demonstrated that the lutein bioavailability from egg is higher than that from other sources such as spinach [47]. This could be explained by the presence of phospholipids in eggs. Indeed, it was shown that a supplement of lutein plus phospholipids was more efficient in increasing plasma lutein concentration than a conventional lutein supplement [48]. Therefore, egg consumption enhanced carotenoid absorption from other carotenoid-rich foods, such as a raw mixed vegetable salad [49]. The food matrix is thus greatly important for carotenoid absorption. Moreover, even if natural sources could not contain as many lutein and/or zeaxanthin levels (ex 10 mg) as food supplements, it does not alter their efficiency due to their high absorption potential. It might also be interesting to explore in future studies other natural sources of carotenoids than eggs.

Considering MPOD, we observed a slight but significant increase in MPOD at all eccentricities for all participants consuming two eggs per day for 4 months (total sample, Table 4, V4 vs. V0). However, there was no further increase in the MPOD value in the enriched group compared with the standard group. This could be explained by the standard egg composition. Standard eggs contained a meaningful amount of lutein/zeaxanthin (~24% of the daily intake observed in the French population [50]) and DHA (~15% of the daily intake observed in the French population [51]) content; thus, due to the presence of phospholipids in eggs, these contents could be sufficient to increase the macular pigment during the four-month period of the study. On the other hand, participants with lower MPOD levels at baseline responded well to lutein supplementation [45]. As the participants of our study were healthy, it was not surprising to observe only a slight increase in MPOD. It is interesting to emphasize that similar data were recently reported by other teams using dietary supplements [52] or eggs [44]. After 6 months of lutein and zeaxanthin dietary supplementation and despite plasma levels showing continuous exposure to lutein and zeaxanthin, MPOD was not modified [52]. In the same way, the consumption of one egg per day enriched with lutein and zeaxanthin for 8 weeks did not affect macular pigment levels, although an increase in serum lutein and zeaxanthin was observed [44]. These results could be explained by a high MPOD level at baseline, limiting the ability to increase further with supplementation. However, based on secondary exploratory analyses of the AREDS2 study, it is actually well accepted that the use of lutein/zeaxanthin supplements is beneficial in reducing the risk of AMD progression [53]. A study of patients over a 15-year period also showed that a moderate consumption of eggs can significantly reduce a patient’s risk of developing incident, late-stage age-related macular degeneration [54].

The main originality of our study was the correlation observed between MPOD at all eccentricities (0.5°, 1°, 2°, and 4°) and lutein concentration in HDL. To our knowledge, this is the first report of this correlation. Some studies have reported an association between MPOD and serum lutein and/or zeaxanthin levels [55,56,57]. However, as in our study, this correlation was not observed in all findings [58,59]. Due to the importance of HDL in lutein transport [18], lutein concentration in HDL may thus be a novel biological marker and correlate of MPOD after lutein supplementation.

In this study, whereas plasma triglycerides were unchanged, the levels of total cholesterol, HDL cholesterol, and LDL cholesterol were slightly but significantly increased in the whole sample, thus suggesting that a daily consumption of two eggs for four months would affect cholesterol metabolism. Interestingly, the LDL/HDL ratio was not significantly increased. For a few decades, common beliefs have emerged on the relationship between egg consumption and health. Attention was first drawn to the cholesterol content of eggs through studies suggesting that cholesterol-rich food consumption may elevate blood cholesterol and increase the risk of coronary heart disease [60]. However, recent trials have revealed that the regular consumption of eggs is not associated with an increased risk of coronary heart disease [55,61,62,63,64], although heavy egg consumption over a longer period of time requires studies to point out possible adverse effects. Considering our study results, daily egg consumption may be something to consider.

In the current trial, a decrease in blood pressure was observed for all participants consuming two eggs per day for 4 months. This decrease could be due to the protein content of eggs. Indeed, Teunissen-Beekman and collaborators showed in 2012 that blood pressure can be lowered by exchanging the intake of carbohydrates with an increased intake of proteins in the context of an isocaloric weight-maintaining diet [65]. Additionally, a change in plasma fatty acid and lipoprotein profile towards the anti-atherogenic range observed in the group consuming DHA-enriched eggs (i.e., higher plasma DHA, HDLC, without any change in plasma triglycerides and balanced LDL/HDL ratio) could contribute to improved endothelial reactivity and lower diastolic blood pressure, which is consistent with observations from large randomized cardiovascular prevention studies [66].

One of the limitations of the current study is the methodology used to evaluate MPOD. Despite the great increase in serum lutein/zeaxanthin in participants consuming enriched eggs for 4 months, the increase in MPOD was limited. A recent meta-analysis revealed that lutein, zeaxanthin, and meso-zeaxanthin supplementation improved MPOD in both AMD patients and healthy participants in a dose-dependent manner [67]. However, although we observed an increase in MPOD in the enriched group, we expected higher MPOD values than those observed in the standard group by using a modified confocal scanning laser ophthalmoscope (SLO, model mpHRA, Heidelberg Engineering, Heidelberg, Germany), a highly reproductible method [68]. Thus, regarding the experimental results, we could consider that the methodology used to evaluate MPOD values in this study was not optimal. The most suitable method for measuring macular pigment was not clearly identified [69]. Scientific debate is still open; numerous studies have compared different methods and investigated the advantages and limitations of measurement techniques [70,71,72]. Another limitation of our study could be the age of the sample. We speculate that young people may not be a useful group for evaluating the effect of dietary supplementation on preserving macular health. An older sample might have responded differently to dietary supplementation. Moreover, as macular pigment is relatively stable over a long period, greater study duration could have allowed the observation of a higher MPOD increase. Nevertheless, the study protocol was well designed (double-blind, randomized), and compliance with eggs was good (>93% overall). Furthermore, no subject was lost during the trial period (4 months).

## 5. Conclusions

This study showed that the consumption of two eggs (standard or lutein/DHA-enriched) per day for 4 months slightly but significantly increased MPOD at all eccentricities. Plasma lutein, zeaxanthin, and DHA concentrations increased in both groups but to a greater extent in the enriched egg group. The major original result of this trial is the correlation between the MPOD value and HDL lutein concentration observed in participants consuming enriched eggs. However, this correlation should be confirmed by other clinical studies.

## Figures and Tables

**Figure 1 nutrients-13-03347-f001:**
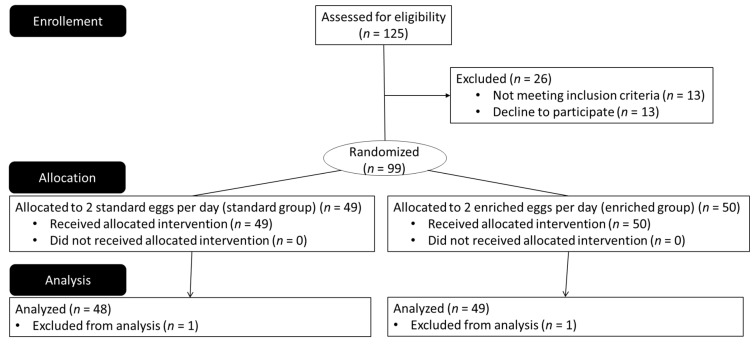
ALGOVUE CONSORT diagram.

**Figure 2 nutrients-13-03347-f002:**
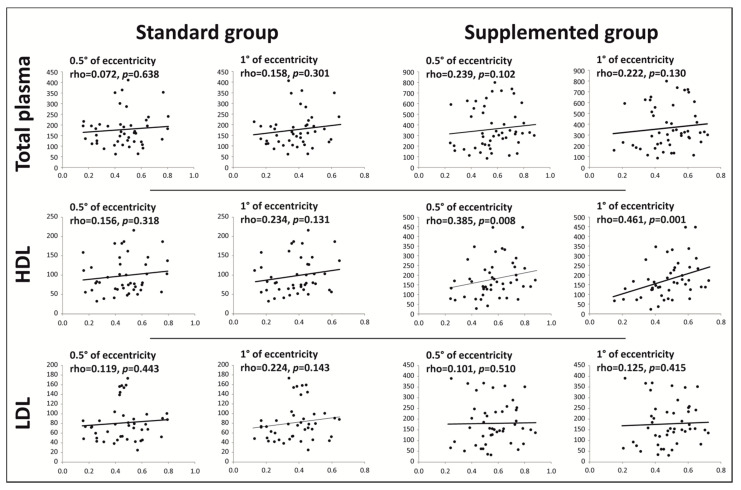
Correlation between total plasma, HDL, and LDL concentrations of lutein and MPOD at 0.5° and 1° eccentricities. Lutein concentrations are expressed as ng/mL of plasma in total plasma and as ng/mg of proteins in HDL and LDL fractions (*y* axis). MPOD is expressed as density units (*x* axis). *n* = 48 in standard group, *n* = 49 in enriched group. Correlations were assessed by using Spearman’s test.

**Table 1 nutrients-13-03347-t001:** Lutein, zeaxanthin, vitamin A, vitamin E, and polyunsaturated fatty acid contents of the eggs used in the study.

Nutrient	Concentration per Egg			Concentration for Two Eggs	
	Standard	Enriched	*p*-Value	Standard Group	Enriched Group
**Lutein, mean (SD), mg**	0.120 (0.013) *	0.961 (0.142) *	**1.64 × 10^−5^**	0.24	1.922
**Zeaxanthin, mean (SD), mg**	0.065 (0.009) *	0.100 (0.012) *	**2.06 × 10^−5^**	0.13	0.2
**DHA, mean (SD), mg**	37.6 (6.5) *	134. 5 (14.7) *	**1.64 × 10^−5^**	75.3	268.9
**EPA, mean, mg**	-	-	-	-	-
**LA, mean, mg**	479.9	440. 8	-	959.9	881. 6
**ALA, mean, mg**	11.2	45.2	-	22.3	90.4
**Vitamin A, mean (SD), µg**	0.178	0.224	-	0.356	0.448
**Vitamin E, mean (SD), µg**	1.40	1.57	-	2.8	3.14

Abbreviations: ALA = alpha-linolenic acid, DHA = docosahexaenoic acid, EPA = eicosapentaenoic acid, LA = linolenic acid, SD = standard deviation. * *n* = 13, statistical analyses were performed using paired *t* tests. For other values, no statistical test was performed.

**Table 2 nutrients-13-03347-t002:** Baseline characteristics of the participants.

Baseline Characteristics	Standard Group (*n* = 49)	Enriched Group (*n* = 50)
Gender		
Female	25	24
Male	24	26
Age, mean (interquartile range), year	29.8 (23–36)	32.8 (24–40)
BMI, mean (SD), kg/m^2^	21.58 (2.99)	22.13 (2.17)
Blood pressure, mean (SD), mm Hg		
Systolic	119.6 (8.90)	121.9 (10.03)
Diastolic	66.97 (7.13)	67.70 (6.81)
Heart rate, mean (SD), bpm	62.91 (6.44)	63.60 (4.04
Triglycerides, mean (SD), g/L	0.68 (0.30)	0.77 (0.22)
TC, mean (SD), g/L	1.74 (0.27)	1.88 (0.29)
HDL-C, mean (SD), g/L	0.57 (0.11)	0.58 (0.13)
LDL-C, mean (SD), g/L	1.03 (0.25)	1.14 (0.26)
LDL-C/HDL-C, mean (SD)	1.90 (0.66)	2.07 (0.65)
Xanthophyll carotenoids, mean (SD), μg/mL		
Lutein	154.4 (62.7)	167.4 (84.3)
Zeaxanthin	17.6 (9.0)	17.7 (10.4)
DHA, mean (SD), mole % of total FA		
Plasma	1.94 (0.65)	1.89 (0.53)
Erythrocytes	1.84 (0.94)	2.04 (0.93)
MPOD, mean (SD)		
0.5°	0.551 (0.20)	0.562 (0.15)
1°	0.459 (0.15)	0.474 (0.13)
2°	0.255 (0.09)	0.260 (0.08)
4°	0.113 (0.045)	0.113 (0.040)

Abbreviations: SD = standard deviation, V0 = Visit 0, V4 = Visit 4, BMI = body mass index, bpm = beat per minute, TC = total cholesterol, HDL-C = high-density lipoprotein cholesterol, LDL-C = low-density lipoprotein cholesterol, FA = fatty acid, DHA = docosahexaenoic acid, EPA = eicosapentaenoic acid, MPOD = macular pigment optical density.

**Table 3 nutrients-13-03347-t003:** Clinical parameters of participants before and after daily consumption of two standard or lutein-zeaxanthin-DHA-enriched eggs for 4 months.

	Total Sample (*n* = 97)			Standard Group (*n* = 48)			Enriched Group (*n* = 49)			Comparison of Mean Change between V0 and V4 in Enriched Group vs. Standard Group
	V0	V4	*p*-Value ^(1)^	V0	V4	*p*-Value ^(1)^	V0	V4	*p*-Value ^(1)^	*p*-Value ^(2)^
BMI, mean (SD), kg/m^2^	21.85 (2.61)	21.97 (2.63)	0.14	21.58 (2.99)	21.52 (2.92)	0.66	22.13 (2.17)	22.42 (2.25)	**0.004**	**0.031**
Blood Pressure, mean (SD), mm Hg										
Systolic	120.7 (9.51)	118.4 (8.71)	**0.005**	119.6 (8.90)	116.6 (8.74)	**0.023**	121.9 (10.03)	120.1 (8.41)	0.08	0.55
Diastolic	67.34 (6.94)	63.95 (6.67)	**2.40 × 10^−5^**	66.97 (7.13)	63.18 (6.67)	**0.003**	67.70 (6.81)	64.72 (6.65)	**0.001**	0.59
Heart Rate, mean (SD), bpm	63.26 (5.36)	62.84 (4.25)	0.34	62.91 (6.44)	62.47 (4.51)	0.37	63.60 (4.04	63.20 (3.99)	0.77	0.36
Triglycerides, mean (SD), g/L	0.72 (0.26)	0.74 (0.34)	0.92	0.68 (0.30)	0.68 (0.27)	0.73	0.77 (0.22)	0.80 (0.39)	0.84	0.70
TC, mean (SD), g/L	1.81 (0.29)	1.90 (0.31)	**6.00 × 10^−5^**	1.74 (0.27)	1.80 (0.25)	**0.043**	1.88 (0.29)	2.00 (0.33)	**0.0004**	0.14
HDL-C, mean (SD), g/L	0.57 (0.12)	0.59 (0.12)	**0.013**	0.57 (0.11)	0.58 (0.10)	0.17	0.58 (0.13)	0.60 (0.14)	**0.03**	0.61
LDL-C, mean (SD), g/L	1.09 (0.26)	1.15 (0.27)	**0.0002**	1.03 (0.25)	1.08 (0.21)	0.053	1.14 (0.26)	1.23 (0.29)	**0.001**	0.17
LDL-C/HDL-C, mean (SD)	1.99 (0.66)	2.03 (0.63)	0.054	1.90 (0.66)	1.89 (0.50)	0.42	2.07 (0.65)	2.16 (0.71)	**0.049**	0.47

Abbreviations: SD = standard deviation, V0 = Visit 0, V4 = Visit 4, BMI = body mass index, bpm = beat per minute, TC = total cholesterol, HDL-C = high-density lipoprotein cholesterol, LDL-C = low-density lipoprotein cholesterol. ^(1)^ The change between V0 and V4 was tested using a paired *t* test. ^(2)^ The effect of the supplementation was evaluated by comparing the mean change in each group using the *t* test.

**Table 4 nutrients-13-03347-t004:** MPOD and plasmatic xanthophyll and DHA concentrations in participants before and after daily consumption of two standard or lutein-zeaxanthin-DHA-enriched eggs for 4 months.

	Total Sample (*n* = 97)			Standard Group (*n* = 48)			Enriched Group (*n* = 49)			Comparison of Mean Change between V0 and V4 in Enriched Group vs. Standard Group
	V0	V4	*p*-Value ^(1)^	V0	V4	*p*-Value ^(1)^	V0	V4	*p*-Value ^(1)^	*p*-Value ^(2)^
**MPOD, mean (SD), density units**										
0.5°	0.557 (0.18)	0.564 (0.18)	**0.03**	0.551 (0.20)	0.558 (0.20)	0.11	0.562 (0.15)	0.569 (0.16)	0.14	0.98
1°	0.466 (0.14)	0.475 (0.14)	**0.0008**	0.459 (0.15)	0.467 (0.15)	**0.01**	0.474 (0.13)	0.483 (0.13)	**0.025**	0.8
2°	0.258 (0.08)	0.265 (0.08)	**0.0005**	0.255 (0.09)	0.264 (0.09)	**0.0008**	0.260 (0.08)	0.266 (0.08)	0.079	0.4
4°	0.113 (0.04)	0.117 (0.04)	**0.0028**	0.113 (0.045)	0.119 (0.04)	**0.0013**	0.113 (0.040)	0.115 (0.04)	0.22	0.19
**Lutein, mean (SD)**										
Total plasma, ng/mL	160.9 (74.2)	273.7 (177.1)	**1.17 × 10^−14^**	154.4 (62.7)	178.1 (76.78)	**0.001**	167.4 (84.3)	369.3 (197.1)	**3.13** **× 10^−16^**	**5.26 × 10^−13^**
HDL fraction, ng/mg proteins	89.44 (46.27)	139.73 (87.07)	**1.15 × 10^−10^**	87.41 (37.86)	99.31 (47.35)	**0.02**	91.44 (53.60)	179.26 (98.69)	**4.37 × 10^−11^**	**1.30 × 10^−8^**
LDL fraction. ng/mg proteins	84.71 (44.60)	131.84 (90.81)	**5.60 × 10^−8^**	83.88 (41.18)	82.28 (37.66)	0.90	85.56 (48.31)	182.5 (101.1)	**1.75 × 10^−12^**	**5.36 × 10^−11^**
**Zeaxanthin, mean (SD)**										
Total plasma, ng/mL	17.6 (9.7)	26.0 (14.0)	**1.31 × 10^−15^**	17.6 (9.0)	22.7 (10.76)	**7.13 × 10^−6^**	17.7 (10.4)	29.2 (16.1)	**3.38** **× 10^−13^**	**0.003**
HDL fraction, ng/mg proteins	10.20 (6.34)	14.17 (8.56)	**1.01 × 10^−11^**	10.43 (5.87)	13.32 (8.17)	**5.30 × 10^−4^**	9.97 (6.83)	15.0 (8.95)	**1.00 × 10^−9^**	**0.01**
LDL fraction, ng/mg proteins	9.64 (6.80)	12.47 (8.43)	**1.31 × 10^−6^**	9.74 (6.60)	10.5 (6.42)	0.05	9.53 (7.07)	14.4 (9.76)	**1.50 × 10^−6^**	**0.01**
**DHA, mean (SD), mole % of total FA**										
Total plasma	1.91 (0.59)	2.32 (0.71)	**1.54 × 10^−10^**	1.94 (0.65)	2.07 (0.65)	**0.03**	1.89 (0.53)	2.56 (0.68)	**1.00** **× 10^−11^**	**2.15 × 10^−6^**
Red Blood cells	1.94 (0.93)	3.54 (1.41)	**7.80 × 10^−20^**	1.84 (0.94)	2.97 (1.34)	**3.11 × 10^−7^**	2.04 (0.93)	4.10 (1.24)	**2.71** **× 10^−15^**	**5.15 × 10^−3^**
HDL fraction	2.48 (0.70)	2.90 (0.84)	**1.36 × 10^−8^**	2.48 (0.72)	2.62 (0.75)	0.090	2.47 (0.69)	3.18 (0.84)	**2.62** **× 10^−9^**	**1.26 × 10^−5^**
LDL fraction	1.57 (0.53)	1.81 (0.58)	**3.53 × 10^−5^**	1.59 (0.50)	1.63 (0.53)	0.505	1.56 (0.49)	1.99 (0.59)	**1.49** **× 10^−5^**	**2.70 × 10^−5^**

Abbreviations: SD = standard deviation. V0 = Visit 0, V4 = Visit 4, FA = fatty acid, DHA = docosahexaenoic acid, MPOD = macular pigment optical density. In HDL, values are expressed as the mean ± SD in ng per mg of ApoA1. In LDL, values are expressed as the mean ± SD in ng per mg of ApoB10. Comparison of means was done using the *t* test. ^(1)^ The change between V0 and V4 was tested using a paired *t* test. ^(2)^ The effect of the supplementation was evaluated by comparing the mean change in each group using the *t* test.

## Data Availability

The data presented in this study are available on request from the corresponding author.

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
