# Peer review of "The ALGOVUE Clinical Trial: Effects of the Daily Consumption of Eggs Enriched with Lutein and Docosahexaenoic Acid on Plasma Composition and Macular Pigment Optical Density"

_nutrients, 2021, doi:10.3390/nu13103347_

Round 1

Reviewer 1 Report

The ALGOVUE clinical trial: effects of the daily consumption
of eggs enriched with lutein and docosahexaenoic acid on
plasma composition and macular pigment optical density

Review

I was excited to read the article but was let down by the minimal change or information to scientific literature. The authors findings are of minimal interest. It is known that animals that are fed with carotenoids have enriched profile in food provided by the animal like eggs or meat. 

It is also known that carotenoid when given in greater amount increased serum levels and ocular levels.

In this case they use a natural source than the nutritional supplement.

There are some clarifications needed.

Authors report overall egg consumption compliance. They need to report this separately for each group. 

The egg cooking process was not controlled. How did they account for the type of oils and the extra DHA they may have got?

The increase in cholesterol is concerning. So authors need to comment on this perhaps not daily but alternate days or 4 times per week might be a better schedule for egg consumption.

Given that the standard egg gives a nice increase in serum levels of carotenoids. Why will one need the enriched eggs?

The participants are incredibly healthy which is not a good sample but I am nots sure if this is representative of French population or a random occurrence of chance? The diastolic BP is low... can they compare it the average papulation in France for the age group analyzed?

Authors should expand on the study comparison of nutritional supplementation versus this study. They have started on an elegant comparison Obana et al showed 1.7 times increase at six months.

Numerous studies like Obana has looked at serum levels changes with supplementation. I think its not enough to single out one investigator. Can the authors do more comparison? Bone et al recently looked at changes with nutritional supplement in young group and serum analysis and MPOD for example. compare this and these type of studies. Then it makes a more robust conclusion that natural substances like egg consumption is better than nutritional supplement. 

On the flip side its very difficult to get 10 mg of zeaxanthin or lutein from natural sources like eggs and that will also need to be mentioned. That more robust sustainable diets using natural substances, that does not cause elevation of cholesterol will need to be explored in future studies.

I like the plasm concentrations, HDL and LDL and MPOD levels. The problem is they need to plot the change in MPOD. In its current form I am not sure what clinical conclusions I can draw.

Reviewer 2 Report

Journal: Nutrients

Manuscript ID: nutrients-1332365

Title: The ALGOVUE clinical trial: effects of the daily consumption of eggs
enriched with lutein and docosahexaenoic acid on plasma composition and
macular pigment optical density.

Authors: Coralie Schnebelen-Berthier *, Niyazi Acar, Emilie Simon,
Clémentine Thabuis, Anne Bourdillon, Adeline Mathiaud, Luc Dauchet, Cécile
Delcourt, Pascale Benlain, Martine Crochet, Sabine Defoort, Anne Tailleux,
Bart Staels, Lionel Bretillon, Jean-Michel Lecerf

Reviewers comments

  1. What is the dietary composition of the standard diet and enriched diet?
  2. How accurate is the data generated based on the food questionnaire? It is very difficult to recollect the foods eaten for the last 7 days. Kindly mention its as approximate values in the table. The food contains medium and low level of carotenoids and DHA were consumed by subjects as part of their routine diet. How the baselines are corrected?
  3. HDL is one of the major transport proteins and it transports especially xanthophyll carotenoids from serum to retina. The significant increase in MPOD density doesn’t correlates with HDL serum concentration. Why? The plasma L and Z levels were increased but there is no significant increase in plasma HDL levels and the values are almost the same for the standard group and enriched group?
  4. The egg is not cooked and processed by the subjects according to the standard protocol. The effect of cooking and thermal processing may definitely affect the carotenoid and DHA bioavailability. There will be larger inter-individual variation that may occur among the subjects. The addition of oil, amount and nature of cooking oil, cooking time, thermal processing such as boiling will affect the carotenoids and DHA present in the egg as there are number of factors involved in carotenoid and DHA bioaccessibility and bioavailability.
  5. Did the authors study the carotenoid and DHA concentration in other lipoprotein fractions?
  6. Why the authors preferred normal subjects over AMD subjects?
  7. The daily recommendation of L and Z per day is approximately 6 to 10 mg/day but 2 mg xanthophylls /day in your study is sufficient for the subjects to prevent from ocular diseases? Because higher MPOD values are correlated with healthier eyes.

Round 2

Reviewer 1 Report

Authors have made a good effort to cover the points I have raised.

Author Response

Thank you very much for your comments and suggestions. We are pleased to have been able to meet all your comments.

Reviewer 2 Report

I do not find any major improvement in the revised version. I really appreciate the author's effects though I may reject the paper in the current form. The significance of the study and novelty is lacking.

Author Response

  • It’s true that some information were already known. However, many originalities should be considered :
    • The use of microalgae to enriched eggs,
    • The correlation between the macular pigment optical density (MPOD) value and HDL lutein concentration,
    • The high carotenoid increase in plasma with very low amount of carotenoids in eggs,
    • The simultaneous eggs enrichment in omega-3 fatty acids and carotenoids.